# Rapid Decline of IFN-γ Spot-Forming Cells in Pleural Lymphocytes during Treatment in a Patient with Suspected Tuberculosis Pleurisy

**Osamu Usami [1], Haorile Chagan Yasutan [2,3], Toshio Hattori [3,*] and Yugo Ashino [4,*]**

[1] Department of Respiratory Medicine, Kurihara Central Hospital, Miyagi 987-2205, Japan; usamin@mac.com
[2] Mongolian Psychosomatic Medicine Department, International Mongolian Medicine Hospital of Inner Mongolia, Huhhot 010065, China; haorile@gmail.com
[3] Department of Health Science and Social Welfare, Kibi International University, Okayama 716-0018, Japan
[4] Department of Respiratory Medicine, Sendai City Hospital, Miyagi 982-8502, Japan
[*] Correspondence: hattorit@kiui.ac.jp (T.H.); ya82@yahoo.co.jp (Y.A.); Tel.: +81-866-22-9454 (T.H. & Y.A.)

**Abstract:** A differential diagnosis of tuberculosis pleurisy is often difficult. A 48-year-old Japanese man with no previous medical history visited the outpatient department for dyspnea and fever. His chest-XP and laboratory findings, especially high C-reactive protein levels, indicated pleuritis with pleural effusion. Pleural lymphocytes showed high numbers of spot forming responses in interferon gamma release assay (IGRA). Pleural effusion contained high levels of adenosine deaminase and hyaluronic acid, but no *Mycobacterium tuberculosis* (TB) antigen was detected by culture or polymerase chain reaction (PCR). Although the infectious agent was not detected, the clinical and laboratory findings strongly suggested that he was suffering from tuberculosis pleurisy. After treatment with anti-TB drugs, a rapid decline of spot-forming cells (SFCs) of pleural lymphocyte was observed, despite persistently high levels of other biomarkers and increased pleural lymphocytes. This case demonstrates that an IGRA of pleural lymphocytes would be useful for therapeutic diagnosis for TB pleurisy suspected for TB.

**Keywords:** tuberculosis pleurisy; interferon-gamma release assay; pleural lymphocytes

## 1. Introduction

One of the most important issue in global health is TB infection, and manifestation of TB infection could be affected by changing host immune factors. The increase of diabetes mellitus and immuno-compromised hosts such as human immunodeficiency virus (HIV) infection are the major factors for TB occurrence. TB pleural effusion (TPE) is one of the most common sites of extra-pulmonary TB, although the incidence varies between regions. The incidence of TPE in TB nonendemic areas is 3–5%. In TB endemic areas, however, the incidence approaches 30%, in part due to the high proportion of HIV-positive individuals [1]. Diagnosis of TPE is often difficult because conventional positive rates in pleural fluids are 30% in both PCR and acid-fast bacilli staining [2]. Currently, detection of adenosine deaminase (ADA) in the pleural fluid at a cut-off value of 40 U/L is considered the most reliable test for TPE, with sensitivity and specificity close to 90% [1,3,4]. However, ADA levels are also increased in emphysema and in malignant tumors, along with other conditions, making it an unreliable marker for a definitive TB diagnosis on its own [5]. Hyaluronic acid levels are also known to be elevated in TPE, providing an alternative or combinatorial marker [6]. To overcome these diagnostic difficulties, an interferon-gamma (IFN-γ) release assay (IGRA) such as the QuantiFERON-TB (QFT) test, and enzyme-linked immunospot (ELISPOT) were introduced. However, QFT-GIT test

or its components showed poor accuracy in the diagnosis of TPE, largely because of a high number of indeterminate results due to high background IFN-γ production in the TB pleural effusion [7]. Therefore, IFN-γ itself has been proposed to be a better marker than the IGRA for the diagnosis of TPE [8]. However, it was found that high background was explained by four to five times enrichment of *Mycobacterium tuberculosis* antigen-specific IFN-γ-producing cells in the pleural fluid compared with their levels in the peripheral blood from patients with TPE in parallel assays. Accordingly, an IGRA performed on pleural fluid mononuclear cells could provide an accurate, rapid diagnosis of TPE after proper subtraction of the background signal [9–11]. Until now, it has not been demonstrated if IGRA could reflect the therapeutic response or not. Here, we report a case in which the IGRA (T-Spot) was performed in a patient with suspected TPE, using lymphocytes from the pleural effusion before and after therapy, and demonstrated the effectiveness of this method for the therapeutic diagnosis of TPE.

## 2. Case Presentation Section

A 48-year-old man visited Sendai City Hospital because of dyspnea and left pleural effusion, as defined by the Chest-XP in January 2019 (Figure 1A). We got his informed consent for ethical approval. He had been suffering from fever, fatigue, and a dry cough for one month. His body temperature was 39.2 °C. Mycobacteria staining and culture of sputum were negative due to dry cough. There is no previous TB history. Apparent contact with TB patients was denied. Laboratory findings showed slight anemia and relative neutrophilia with lymphopenia. Increased erythrocyte sedimentation rate and C-reactive protein (CRP) were demonstrated, indicating active inflammation (Table 1). IGRA of peripheral blood lymphocytes was performed by using T-SPOT (Oxford Immunotec, Oxford, UK) and found to be negative. HIV was negative. Both of the two blood culture tests resulted negative. His left respiratory sound was decreased. The computed tomography scan showed a right-shifted trachea with airless left lung lobes and left pleural effusion. Thoracic puncture (day 0) revealed serous effusion (protein; 4.5 g/dL) and high cell numbers 944/μL with 87% lymphocytes without malignant cells. Increased levels of ADA (108 U/L; normal limit 20 U/L), LDH (lactate dehydrogenase) (563 U/L; normal limit 245 U/L), and hyaluronic acid (35,100 ng/mL; normal limit 50 ng/mL) were detected. TB DNA was tested for by PCR (Cobas Taqman 48 Analyzer), using the Cobas Taqman TB kit (Roche Diagnostics), but it was not detected. Furthermore, acid-fast bacteria were tested for by smear and PCR, but they were not detected; and bacterial culture were also negative. IGRA of pleural lymphocytes was performed, using T-SPOT, as described previously ($2.5 \times 10^5$/well) and showed high numbers of SFCs (ESAT-6, 393; CFP-10, 545) [12]. Negative control cells cultured without antigens also showed IFN-γ secretion (41 SFCs) (Figure 2A).

Anti-TB drugs (isoniazid, rifampicin, pyrazinamide, and ethambutol) were administered daily since day eight to the patient in the outpatient department. At day 12 (four days after starting anti-TB drugs), the patient sustained dyspnea, and increased effusion was observed, although CRP levels decreased (Figure 1B). Due to the absence of a definite TB diagnosis, the second thoracic puncture was repeated for further investigation. The levels of ADA increased (114 U/L) and those of LDH (563 U/L) and hyaluronic acid (35,100 ng/mL) decreased by 14–20%. (Figure 2B). In particular, the cell numbers in the pleural fluid increased dramatically (by fourfold to 3844/μL) with lymphocytic preponderance (93%). However, SFCs against ESAT-6 decreased dramatically (137 SFCs; 65% reduction), and those against CFP-10 also decreased (252 SFCs; 54% reduction). SFCs cultured without the antigen also decreased (4 SFCs; 91% reduction). At day 30 (22 days after starting anti-TB drugs), cell numbers increased to 4844/μL with 93% lymphocytes. Interestingly, SFCs against ESAT-6 were not detected, and the response to CFP-10 decreased dramatically (83% reduction). The percentage inhibition in ADA, LDH, and hyaluronic acids was 28%, 36%, and 29%, respectively, at day 30 (22 days after starting anti-TB drugs) (Figure 2B). The patient recovered with decreased effusion and symptoms (Figure 1C,D) at day 30 and 60, suggesting anti-TB drugs were effective.

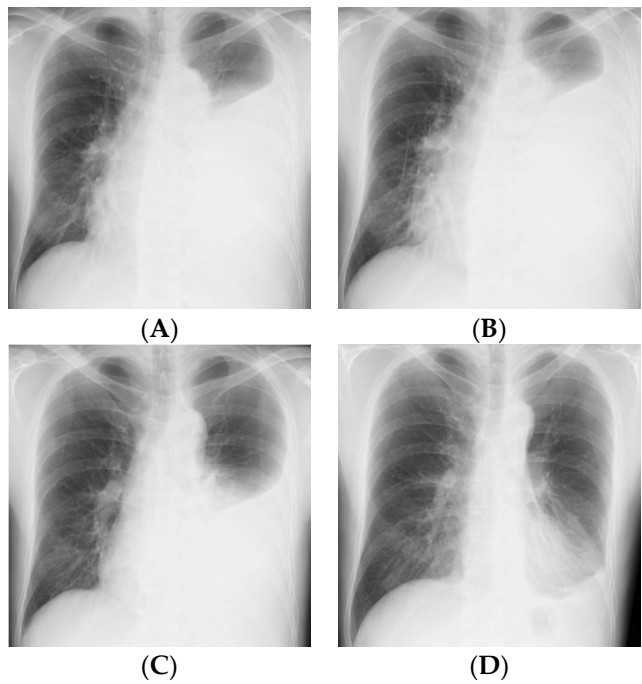

**Figure 1.** Chest X-ray from (**A**) the first visit, (**B**) at day 12, (**C**) at day 30, and (**D**) at day 60.

**Table 1.** Laboratory data.

| Variable (Blood) | Reference Range | Day 0 | Day 12 | Day 30 |
|---|---|---|---|---|
| Hematocrit (%) | 42.0–53 | 37.9 | 42.9 | 42.6 |
| Hemoglobin (g/dL) | 13.5–17.5 | 12.9 | 14.3 | 13.9 |
| White-cell count (per mm$^3$) | 3700–8500 | 8500 | 7600 | 6300 |
| Differential (%) | | | | |
| Neutrophils | 44.0–68.0 | 82.2 | 75.8 | 60.8 |
| Bands | 0.0–10.0 | 0 | 0 | 0 |
| Metamyelocytes | 0 | 0 | 0 | 0 |
| Lymphocytes | 27.0–44.0 | 9.9 | 12.5 | 20 |
| Monocytes | 3.0–12.0 | 7.3 | 6.5 | 6.1 |
| Eosinophils | 0.0–10.0 | 0.5 | 4.7 | 12.5 |
| Basophils | 0.0–3.0 | 0.1 | 0.5 | 0.6 |
| Platelet count (per mm$^3$) | 150,000–3,550,000 | 472,000 | 529,000 | 317,000 |
| Red-cell count (per mm$^3$) | 3,900,000–5,300,000 | 3,990,000 | 4,530,000 | 4,580,000 |
| Urea nitrogen (mg/dL) | 2–80 | 11 | 15 | 8 |
| Creatinine (mg/dL) | 0.65–1.07 | 0.86 | 0.87 | 0.82 |
| Glucose (mg/dL) | 60–110 | 136 | N/A | N/A |
| Alanine aminotransferase (U/liter) | 3–40 | 32 | 21 | 14 |
| Aspartate aminotransferase (U/liter) | 8–35 | 34 | 44 | 18 |
| Protein (mg/dL) | 6.6–8.4 | 6.0 | 6.4 | 6.7 |
| Albumin (mg/dL) | 3.8–5.2 | 2.9 | 3.1 | 3.7 |
| Erythrocyte sedimentation rate(mm/h) | 2–10 | 53 | 40 | 20 |
| C-reactive protein (mg/liter) | 0.00–0.3 | 8.15 | 2.31 | 1.47 |
| Prothrombin time (s) | 10–13 | 11.2 | 11 | 10.9 |
| PT (INR) | 70.0–110.0 | 107.9 | 95 | 114.5 |
| APTT (s) | 23.0–38.0 | 33.4 | 26.4 | 25.8 |
| D-Dimer (ng/mL) | 0.00–1.00 | 2.44 | 11.82 | 8.31 |

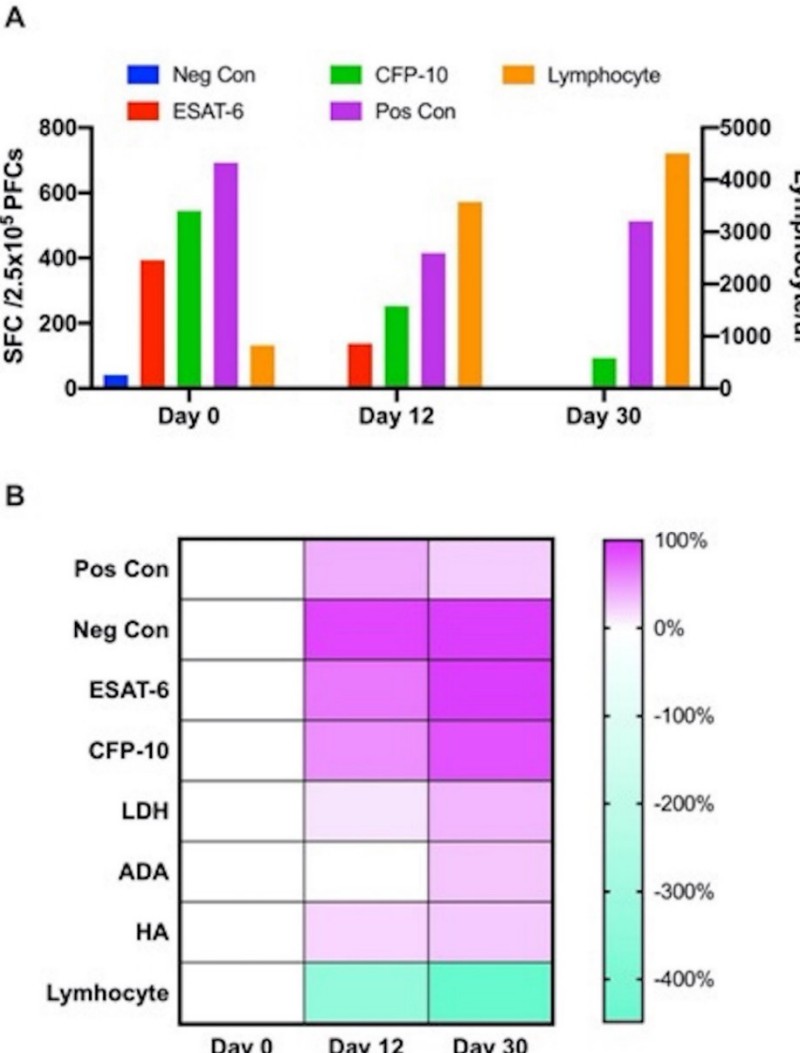

**Figure 2.** Time-dependent changes in biomarkers from the pleural fluid. (**A**) Spot-forming cells (SFCs) of pleural lymphocytes in the T-spot assay. Left *y* axis shows the SFC number of negative control (Neg Con), ESAT-6, CFP-10, and positive control (Pos Con), whereas the right *y* axis shows the number of lymphocytes. (**B**) Heat map of the percent decrease in the expression level of each marker. The percent inhibition is presented in colors ranging from green to pink, as shown in the key, and was calculated as ((Day 0–Day X)/Day 0)) × 100%. Day 0 is the first day of visit. Day 12 indicates four days after the anti-TB drugs were started. Day 30 was 22 days after the anti-TB drugs were started. The results were from the mean of duplicate assay.

## 3. Discussion

In the present case, a clinical diagnosis of TPE was made based on high ADA, LDH, and hyaluronic acid levels, and on the high numbers of SFCs in the T-spot test (IGRA). Lymphocytes of the pleural effusion showed high SFCs in the presence of TB specific antigens, such as ESAT-6 and CFP-10. It is striking that these antigen-specific SFCs in pleural effusion disappeared after four days of treatment, although other markers remained at high levels. Despite the marked decrease of antigen-specific cells, absolute counts of lymphocyte in the pleural fluid increased at four days after therapy. To our knowledge, this is the first case demonstrating IGRA could be a therapeutic diagnostic tool for TB pleurisy.

It was previously reported that ELISPOT assay using pleural fluid showed higher sensitivity and specificity than those of ELISPOT using peripheral blood lymphocytes or ADA of pleural fluid [9,10].

Very recently, ELISPOT of pleural fluids was integrated into the strategy for a more effective diagnosis, especially in differentiating from malignant pleural effusion [11].

It was claimed that QFT tests have poor accuracy in the diagnosis of TPE, largely because of a high number of indeterminate results due to high background IFN-γ production in the TPE [7].

Our study also demonstrated significant numbers of SFCs without antigen simulation; however, they were not detected after therapy. Polyfunctional Th1-cytokine-producing cells were reported in pleural lymphocytes in TPE [13]. Therefore, optimization of ELISPOT conditions would be necessary for examples cut off value of SFCs were determined to be higher than those of peripheral blood lymphocytes [9]. Alternatively, fewer number of pleural fluid lymphocytes ($1 \times 10^5$) were added to each well as compared with cell numbers ($2.5 \times 10^5$)) of peripheral blood lymphocytes [11]. We have also shown the high SFCs in negative control at day 0, because we also used $2.5 \times 10^5$) cells/well. It might be desirable to use different cell numbers for T-spot for optimization for pleural lymphocytes. Therefore, careful isolation of lymphocytes from pleural fluids and optimization of the conditions would be necessary to intrude ELISPOT to the study of pleural lymphocytes.

We previously reported that the level of galectin-9 (Gal-9), an apoptosis-inducing factor, is elevated in TPE, and that treatment of pleural lymphocytes with Gal-9 without antigen stimulation resulted in IFN-γ production, accompanied by apoptosis [14]. Thus, Gal-9 levels might have increased in this patient before therapy and may have decreased after therapy. This decrease might be associated with a lack of lymphocyte apoptosis and may have caused an increase in the pleural lymphocyte cell numbers. Further studies on the relationships of Gal-9 and IGRA in TPE would be necessary in the future.

Finally, our study demonstrated that IGRA using lymphocytes isolated from pleural effusion could be a powerful tool for therapeutic diagnosis of tuberculosis. However, it is necessary to analyze a large number of cases in the future to determine which patients will receive multiple pleural effusion IGRA.

**Author Contributions:** Conceived and designed the study and contributed to the treatment planning, Y.A.; acquired the laboratory and clinical data and wrote the manuscript, O.U. and H.C.Y.; H.C.Y. previous work place was Mongolian Psychosomatic Medicine Department, International Mongolian Medicine Hospital of Inner Mongolia, Hohhot, China; and T.H. made critical revisions, and engaged in review and editing. All authors participated in the multidisciplinary conference and approved the final version of the manuscript.

**Funding:** This research was funded by the Research Program on Emerging and Re-Emerging Infectious Diseases from the Japan Agency for Medical Research and Development (AMED JP18fk0108042h0002).

**Conflicts of Interest:** The authors declare no conflicts of interest.

## Abbreviations

LDH    Lactate dehydrogenase
ADA    Adenosine deaminase
HA     Hyaluronic acid

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
