# Peer review of "Rapid Decline of IFN-γ Spot-Forming Cells in Pleural Lymphocytes during Treatment in a Patient with Suspected Tuberculosis Pleurisy"

_reports, doi:10.3390/reports2040027_

Round 1
Reviewer 1 Report
Rapid decline of the INF- γ spot forming cells in pleural lymphocytes during treatment in patients with suspected tuberculosis pleurisy
Dear author and editor:
This is an interesting case report focused on an important problem in the differential diagnosis of the tuberculosis pleurisy, and suggested the IGRA test of the pleural lymphocytes as indicator for therapeutic diagnosis of TB pleurisy.
This case report could be published in reports after a minor revision:
You mentioned that the patient did not have a previous TB history means no LTB, did you confirm this by Quantiferon test? Did you use the IGRA test also on the peripheral blood? At the end of the discussion, i suggest to write your future recommendation and conclusion to consider or not this test for therapeutic diagnosis. Further studies are need in the future.
Author Response
Point 1: You mentioned that the patient did not have a previous TB history means no LTB, did you confirm this by Quantiferon test? Did you use the IGRA test also on the peripheral blood?
Response 1: IGRA test was done to confirm no previous TB history (line 63).
Point 2: At the end of the discussion, i suggest to write your future recommendation and conclusion to consider or not this test for therapeutic diagnosis. Further studies are need in the future.
Response 2: We described our recommendation and conclusion in the discussion (line 146).
Reviewer 2 Report
In this manuscript, authors reported the case of a 48 years old Japanese man with chest-XP presenting pleuritic with pleural effusion. Clinical and laboratory observations suggested a tuberculosis pleurisy. Authors discussed the idea to use interferon gamma release assay of pleural lymphocytes for therapeutic diagnosis of suspected tuberculosis pleurisy.
The manuscript present quality, novelty and is of interest to the readers. I suggest to accept the manuscript in this current form after minor modifications.
The following suggestions need to be considered:
If available, authors should add data of different time points post antibiotic treatment in table 1. Figure 1: Add chest-X-ray at day 30 to be homogenous with results in figure 2. Figure 2A: Are technical replicates available for this experiment. Is it possible to add standard deviation on the graph? Figure 2B: Could authors comment why neg control present huge increase at day 12 and 30?Minor:
I suggest to use “Mtb” to abbreviate “Mycobacterium tuberculosis” and “TB” for “tuberculosis”. If authors write “Mycobacterium tuberculosis”, do not forget to use italic (line 47). Remove the dot before brackets (line 63).Author Response
Point 1: If available, authors should add data of different time points post antibiotic treatment in table 1.
Response 1: We added day 12 and 30 in table 1.
Point 2: Figure 1: Add chest-X-ray at day 30 to be homogenous with results in figure 2.
Response 2: We added chest X-ray at day 30 in figure 2.
Point 3: Figure 2A: Are technical replicates available for this experiment. Is it possible to add standard deviation on the graph?
Response 3: We described that the results were from the mean of duplicate assay in Figure 2A legend.
Point 4: Figure 2B: Could authors comment why neg control present huge increase at day 12 and 30?
Point 5: I suggest to use “Mtb” to abbreviate “Mycobacterium tuberculosis” and “TB” for “tuberculosis”. If authors write “Mycobacterium tuberculosis”, do not forget to use italic (line 47). Remove the dot before brackets (line 63).
Response 5: We corrected “Mycobacterium tuberculosis” to italic (line 47) and removed the dot before brackets (line 63).